# Of Soldiers and Their Ghosts: Are We Ready for a Review of PTSD Evidence?

Adonis Sfera [1,*], Jonathan J. Anton [2], Hassan Imran [1], Zisis Kozlakidis [3], Carolina Klein [4] and Carolina Osorio [1]

1 California Department of State Hospitals, Patton, CA 92369, USA; carolina.osorio@dsh.ca.gov (C.O.)
2 Department of Biology, California Baptist University, Riverside, CA 92504, USA; jonathan.anton@dsh.ca.gov
3 International Agency for Research on Cancer, World Health Organization, 69366 Lyon, France; kozlakidisz@iarc.who.int
4 California Department of State Hospitals, Sacramento, CA 95814, USA; carolina.klein@dsh.ca.gov
* Correspondence: adonis.sfera@dsh.ca.gov

**Abstract:** Psychosocial trauma has accompanied mankind since time immemorial and has been sufficiently portrayed in art and literature to suggest that posttraumatic stress disorder may be as old as combat itself. Since war is more frequent in human history than peace, public health measures are confined to mitigating the detrimental impact of battlefield experiences on combat participants. At present, PTSD outcome studies show mixed results, marked by high nonresponse rates, therapy dropout, and completed suicide, suggesting that novel strategies are urgently needed. Those of us who work routinely with combat veterans have noted an increasing trend of patients preferring mindfulness-based therapies as opposed to trauma-centered treatments, such as prolonged exposure or trauma-focused cognitive behavioral therapy. Preference for mindfulness over trauma-based therapies appears to coincide with the shift in research focus from the amygdala and fear to the insular cortex and interoceptive awareness. Therefore, rethinking PTSD as insular pathology is driven by the recent findings that neurons in this cortical area not only regulate cardiac rhythm but also record past intestinal inflammations. These discoveries likely explain the high comorbidity of stress-related disorders with premature endothelial senescence and a dysfunctional intestinal barrier. Moreover, the identification of the cholinergic anti-inflammatory pathway and the revelation that endothelial cells express alpha-7 nicotinic receptors has brought PTSD prevention and early detection within reach. In this narrative review, we discuss the relationship between early vascular aging, gut barrier disruption, and PTSD. We also examine the link between this pathology and faulty interoceptive awareness, surmising that hypertension and decreased heart rate variability are PTSD risk factors, while lipopolysaccharide, lipopolysaccharide binding protein, soluble CD14, microbial cell-free DNA, acyloxyacyl hydrolase, and IL22 comprise early detection markers of this disorder.

**Keywords:** post traumatic stress disorder; hypertension; heart rate; microbial translocation; endothelial cells

## 1. Introduction

Although the term "shell shock" was coined during World War I, battle trauma had likely entered human history long before modern warfare [1]. Ancient civilizations, unacquainted with the inner dimensions of the human mind, likely experienced war trauma as a mythical destiny based on the "rule of blood", traced in Ancient Greece to the House of Atreus, a primitive jurisprudence predating the "rule of law" [2,3].

"Nothing Matters to Me Now, But killing and blood and men in agony", exclaimed Trojan hero Achilles, suggesting that experiencing combat can beget new and often random acts of violence (Iliad 19.226). Indeed, combat scenarios, relived over and over by many contemporary veterans, appear to hijack selective neuronal assemblies implicated in emotional recall, keeping the traumatic event vividly alive. Although posttraumatic stress disorder (PTSD) patients maintain awareness of the subjective nature of their adversity-induced

experiences, frequent misinterpretation of innocuous cues as danger signals likely reflects subtle defects of insight. In the section "PTSD and interoceptive awareness", we discuss "dysgnosia", a trauma-mediated partial agnosia, also known as "cognitive bias", a less accurate term unreflective of insight.

In the US, the lifetime prevalence of PTSD in the general population is in the range of 7–8%; however, it can surge to 14–16% in combat veterans and military personnel [4]. These data suggest that stress alone, without individual vulnerability, is insufficient to engender this pathology [4]. On the other hand, in susceptible individuals, the exposure to real or threatened death, severe injury, or sexual assault may be followed by pathology, manifested by nightmares, dissociative phenomena, hypervigilance, avoidance of situations reminiscent of the traumatic experience, and, in a subset of veterans, aggressive behavior directed at the self or others [5,6]. The question we ask here is what constitutes susceptibility, and can the high-risk phenotypes be identified early?

The common psychotherapeutic interventions for PTSD (VA/DoD Clinical Practice Guidelines revised in 2017) are primarily "amygdala-centered" and aim at correcting the cognitive bias, reducing fear associated with the traumatic scenario [7,8]. These approaches include prolonged exposure (PE), cognitive processing therapy (CPT), and trauma-focused cognitive behavioral therapy (CBT) with or without psychopharmacological interventions [9–11]. Although for many patients, these approaches are beneficial, nonresponse and treatment dropout rates are high, while the rate of suicide, the most concerning PTSD manifestation, is around 14%, compared to the lifetime risk of 3.7% in patients with major depressive disorder (MDD) [12]. Some clinicians have noticed that interventions incorporating exposure may sometimes trigger flashbacks, possibly accounting for the high therapy dropout rates [13]. In addition, vivid dreams, common adverse effects of antidepressant drugs, exacerbate nightmares in many patients, contributing to medication non-adherence [14].

Mindfulness-based therapies have emerged as alternative interventions for PTSD, aimed at enhancing interoceptive awareness by promoting conscious living in the "present moment" [15]. Preferred by many patients, mindfulness-based approaches have shown significantly lower dropout rates and have demonstrated statistically significant improvement on self-report [16,17]. This may be significant as the insular cortex (IC) houses a cardiac motor area, linking this neuroanatomical site to the autonomic features of PTSD [18]. For example, stress-induced tachycardia is likely mediated by the IC, while flashbacks and panic attacks may reflect a dysfunction in this neural hub. Moreover, aside from heart rate (HR), the IC has been associated with awareness of error, position of the body and limbs, linking cardiovascular system to insight [19–22].

The communication between the IC and vagus nerve (VN) was defined more than three decades ago, while recent studies have not only confirmed the existence of these pathways, but also found that transcutaneous auricular VN stimulation (taVNS) can enhance interoceptive awareness, suggesting a potential therapeutic strategy for PTSD [23,24]. In addition, the IC is tightly connected to the GI tract and was recently shown to store the memory of prior gut inflammations, likely explaining the high comorbidity between PTSD and inflammatory bowel disease (IBD), estimated at 19.5% [25,26]. Indeed, these findings were validated in neuroimaging studies, revealing impaired insular connectivity in patients with Chron's disease [27,28]. Interestingly, in one study, electrical stimulation of the IC was shown to increase the abundance of gut *Prevotella* and *Bacteroides* species, suggesting that insular neurons control microbiota composition [29].

## 2. PTSD and Interoceptive Awareness

Most PTSD models and therapies assume that excessive activation of amygdala triggers fearful memories, contributing to the pathogenesis of this disorder [30–32]. However, novel studies have shown that the inner perception of trauma may be more significant for the development of posttraumatic symptoms than the activation of fear-imbued memories [32]. For example, when processing traumatic experiences, the IC permeates the

interoceptive and exteroceptive stimuli with autonomic coloring, generating a unique, individualized, inner perception of the catastrophic event [33,34]. In addition, the IC regulates empathy, compassion, and social cues, including the ability to interpret the feelings of others and adjust one's behavior accordingly [35]. This special form of insight is often impaired in PTSD patients, who frequently misinterpret neutral social signals as hostility, leading to hypervigilance, anxiety, and, occasionally, aggression [36]. Furthermore, dissociative phenomena, such as flashbacks and intrusive memories, may reflect disruption of insula-mediated emotional intelligence as opposed to the amygdala and fear [37–39]. Indeed, neuroimaging studies of PTSD patients, experiencing active flashbacks, have revealed both dysconnectivity and impaired insular activation [40,41] (Figure 1).

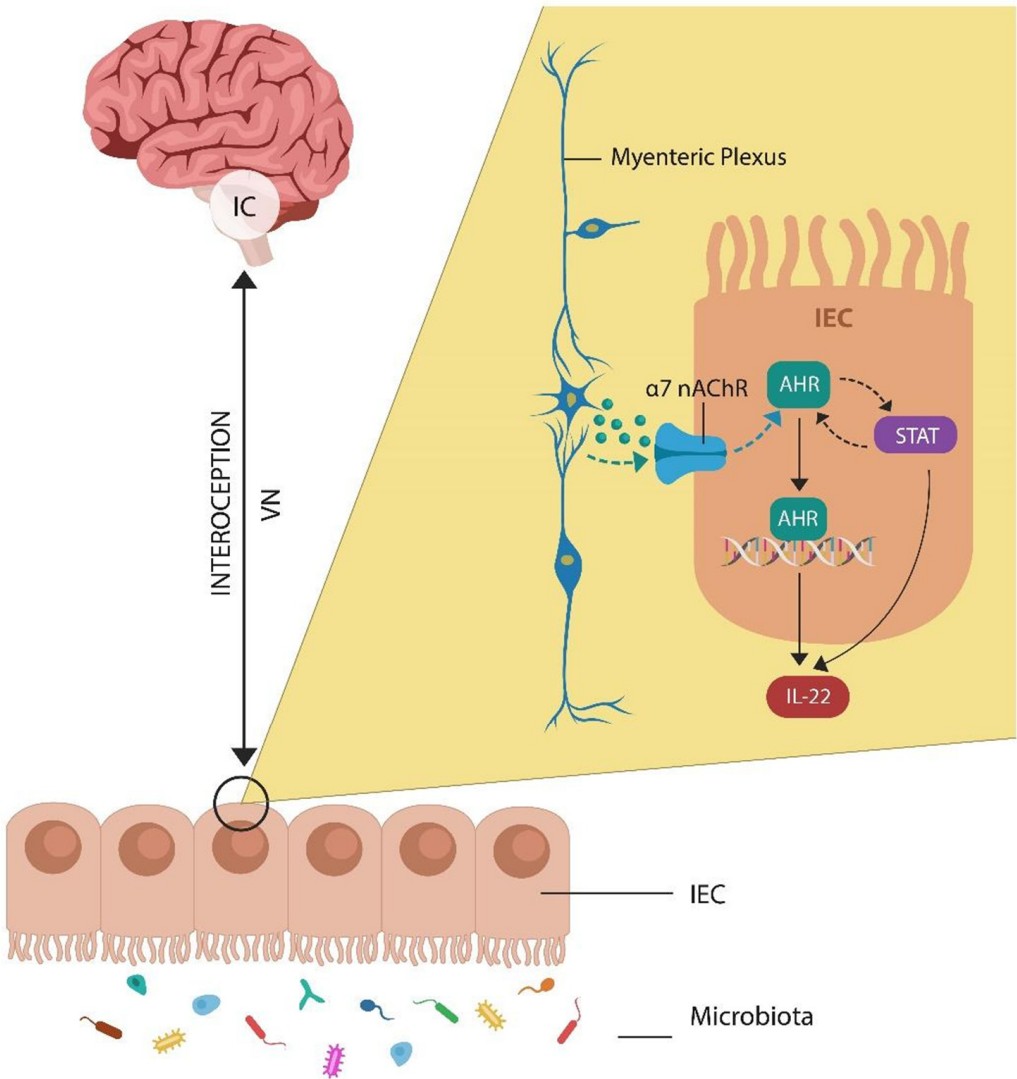

**Figure 1.** The IC-IEC dialog is part of the gut–brain axis as well as the CAP. These systems enable interoceptive awareness, a two-way communication between insular neuronal networks and intestinal epithelial/endothelial cells. At the molecular level, myenteric-plexus-secreted acetylcholine (ACh) (via α7nAChR) activates the AhR/STAT3/IL22 system (which requires AhR entry into the nucleus, where it acts as a transcription factor). IL22, the "guardian of intestinal barrier", prevents MT into host tissues.

Under normal circumstances, the IC receives input from the autonomic nervous system, comprising sympathetic, parasympathetic, and enteric divisions, as well as innate and adaptive immune data, maintaining a detailed inflammatory map [42–44]. In the gut, intestinal epithelial cells (IECs) express alpha 7 nicotinic acetylcholine receptor (α7nAChR),

a major component of the cholinergic anti-inflammatory pathway (CAP) discovered in 2000 by Borovikova LV and colleagues. This system likely explains the beneficial effect of transcutaneous auricular vagal nerve stimulation (taVNS) on the intestinal barrier, where dopamine (DA)-activated aryl hydrocarbon receptor (AhR) upregulates interleukin 22 (IL22), maintaining physiological permeability [45–48] (Figure 1).

Under pathological circumstances, the IC has been implicated in a wide variety of syndromes marked by deficient insight, ranging from anosognosia to psychopathy, and frontotemporal dementia (FTD), linking insular neurons to disease awareness [19,49,50]. This is significant as several viral infections highly comorbid with PTSD, including SARS-CoV-2 and human immunodeficiency virus 1 (HIV-1), disrupt IC, inducing anosognosia for cognitive deficits. This suggests that insula is the likely the "dwelling place of insight" [51–53]. Moreover, as these viral infections induce intestinal inflammation, the GI tract likely plays a role in interoceptive awareness, justifying the vernacular expression "gut feeling", as a synonym for insight [26,27,54]. This is significant as patients with PTSD likely exhibit dysgnosia, a subtle insight deficit, interfering with the appreciation of situational reality [55,56]. For example, interpreting innocuous stimuli as danger signals may represent errors of insight, rather than amygdala-generated fearful stimuli [57]. In addition, like in frontotemporal dementia, PTSD patients may engage in aggressive acts while maintaining some degree of insight into the negative impact of this behavior [58]. Moreover, the fact that some patients with PTSD prefer mindfulness, instead of trauma-oriented therapies, may be the proof of concept that a dysfunctional IC may drive the pathogenesis of PTSD [13,59,60].

In the following subsections, we will discuss in more detail the role of the IC in emotional intelligence, cardiac, and gut connection, as well as PTSD risk factors, early detection markers, and public health approaches for preventing microbial translocation (MT) into the host circulation.

### 2.1. Heart Rate Variability as a PTSD Risk Factor

In colloquial expression, the brain is the epitome of rational thinking and intelligence, while the heart is generally associated with feelings and emotion. Over the past two decades, the term "emotional intelligence" has been circulated widely, although the neuroanatomic "home" of this social skill has yet to be defined. Here, we suggest that social or emotional intelligence dwells in the IC and may be a function of von Economo neurons (VENs), which populate the anterior insula and the cingulate cortex [61–63]. Indeed, dysfunctional VENs have been associated with suicidal behavior and psychosis, while FTD is characterized by the selective loss of these cells [64]. Aside from neuropathology, VENs have been implicated in cardiac physiology as they regulate HR and blood pressure (BP), parameters associated with both emotional expression and environmental demands [65,66]. Indeed, VENs may comprise the heart motor center, which was recently identified in the IC, and plays a major role in the autonomic accompaniment of emotional experiences [67]. The IC communicates with the heart via VN and postganglionic cardiac neurons, likely explaining heart rate variability (HRV) in response to inner or outer stimuli, such as emotions or the environment [67]. Moreover, GI tract inflammation has been associated with decreased HRV, suggesting IC regulation of the autonomic nervous system [68]. Interestingly, dysfunctional resting HR as well as decreased HRV have been documented in psychopathy, a pathology associated with insular damage [69–71]. This may explain the evolutionary preservation of psychopathy, a phenotype likely conserved for its battlefield advantages.

Since under normal circumstances, VENs drive both empathy and HRV, in pathological conditions, these cells may trigger anxiety attacks, hypervigilance, avoidant behavior, or aggression [63,72,73]. HRV was previously identified as a PTSD risk factor, emphasizing the key role of the cardiovascular system in this pathology [74–76].

### 2.2. PTSD-Associated Hypertension (HTN)

Under physiological circumstances, the IC gathers real-time interoceptive and exteroceptive input, permeating it with autonomic hues, memories of past experiences, and

personal values. These data are subsequently communicated to the cardiovascular system via insular–cardiac circuits, generating individualized homeostatic responses [77]. The IC-heart dialog regulates the HR and BP, parameters driven by both emotional experiences and environmental demands [78]. Indeed, the right IC has been associated with HTN, while the left induces vasodilation [79]. BP fluctuations accompany many human experiences, including emotion, physical pain, psychosocial stressors, infections, and pro-survival risk taking [80–82]. For example, the major HTN driver, angiotensin II (ANG II), has been associated with PTSD as renin-angiotensin (RAS) is a well-established component of the human stress response [83,84]. Indeed, ANG II has been associated with the neural expression of fear, while alamandine, a less-studied RAS member, lowers fear by acting on the rostral insula, likely averting the development of PTSD [78,85]. The role of RAS in neuropathology was emphasized during the COVID-19 pandemic, as SARS-CoV-2, the etiologic agent of this viral infection, employs angiotensin-converting enzyme 2 (ACE-2) as the entry portal into host cells.

RAS comprises two arms, the proinflammatory/prooxidative and anti-inflammatory/antioxidative branch, which maintain the homeostasis of both BP and the human stress response. ACE-2 bridges the two RAS branches, indicating that a defect of this enzyme, including viral hijacking, can disable the entire protective RAS branch (Figure 2). This, in return, leads to endothelial barrier disruption and spill-over of neurotoxic molecules, including ANG II or endothelin-1 (ET-1), into the CNS, triggering pathology, including PTSD [86]. Indeed, this mechanism may explain the high comorbidity of some viral infections with both HTN and PTSD [87,88] (Figure 2). In addition, viruses have been known for disrupting both epithelial cells and ECs, increasing the permeability of biological barriers, including the intestinal and blood–brain barrier (BBB), enabling MT into the host systemic circulation [89]. As we have discussed this issue in another article, will not go into more detail here [90].

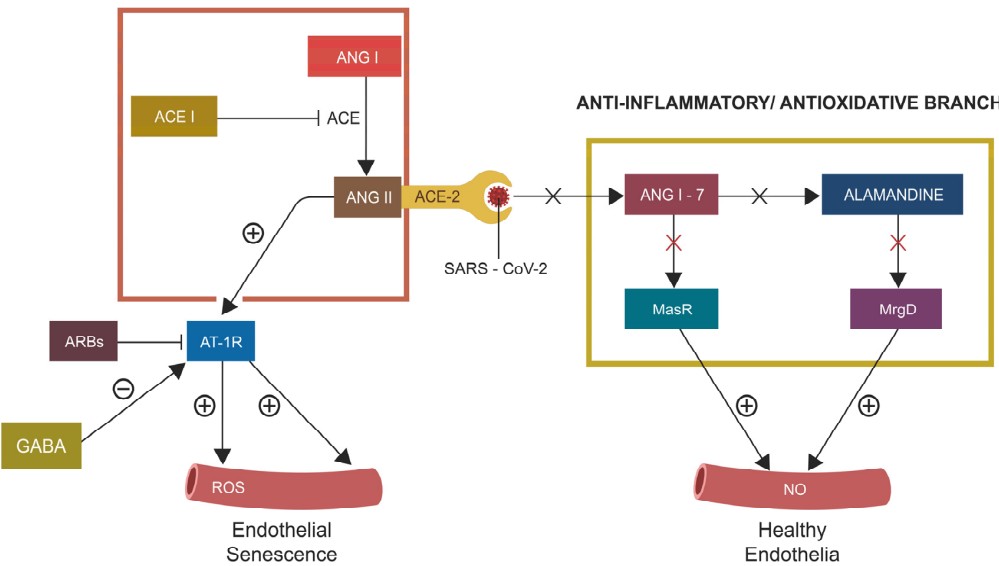

**Figure 2.** The two RAS branches: pro-inflammatory/pro-oxidant and anti-inflammatory/antioxidant, are connected via ACE-2. When ACE-2 is disabled by various pathologies, including the SARS-CoV-2 virus attachment, the entire protective RAS branch is inhibited, disrupting the ECs (comprising the endothelial gut barrier) and enabling MT. ARBs and GABA oppose the effects of ANG II, averting EC senescence, barrier disruption, and MT.

The COVID-19 pandemic emphasized the RAS-aryl hydrocarbon receptor (AhR) link, which plays a key role in orchestrating virus-induced senescence (VIS) in ECs [91–94]. Conversely, angiotensin receptor blockers (ARBs) and γ-aminobutyric acid (GABA) counteract

EC senescence, indicating potential therapeutic effects in PTSD. Indeed, both GABA and ARBs have shown protective effects against psychosocial stress in both human and animal studies [95–97].

In summary, several lines of evidence implicate dysfunctional IC in PTSD pathogenesis, suggesting that impaired HRV and BP comprise risk factors for this disorder. At the molecular level, defective RAS and AhR contribute to PTSD by inducing premature epithelial and endothelial senescence, causing gut barrier disruption and MT. In the following sections, after a brief reminder of AhR physiology and pathology, we discuss PTSD resilience systems, HIF-1α and CAP, that not only counteract MT but also ameliorate IC homeostasis.

### 3. Aryl Hydrocarbon Receptor and Enteric Nervous System: Quick Reminder

The enteric nervous system (ENS), comprising the myenteric and submucosal plexus, is a semi-independent "gut brain" situated within the wall of the GI tract [98]. The ENS contains glial cells and three times as many neurons as the spinal cord. The ENS neurons exhibit unusual plasticity and a high level of neurogenesis as they are replaced every few weeks with newly born neuronal cells [99]. Moreover, the ENS neurons are regulated by AhR, which, depending on the attached ligand, may play a pro- or anti-apoptotic role [100]. Initially, AhR was believed to interact only with the environmental toxin dioxin (2,3,7,8-tetrachlorodibenzo-p-dioxin). However, the discovery that the major neurotransmitters DA, serotonin (5HT), and melatonin, as well as several psychotropic drugs, including clozapine, are AhR ligands, brought this receptor into the neuropsychiatric arena [46,101–104]. Furthermore, when AhR binds to gut indole, it promotes adult neurogenesis in various brain niches, suggesting novel targets for major depressive disorder (MDD) and PTSD [105–108]. It has been well established that AhR regulates corticotropin releasing hormone (CRH), a major PTSD driver, involved in interoceptive awareness and the IC [109,110].

### 3.1. AhR and Cellular Senescence

During development, AhR plays a key role in organogenesis; however, excessive activation of this transcription factor later in life has been associated with premature cellular senescence and neurodegeneration [111,112]. AhR is a transcription factor, stabilized in the cytosol by the heat shock protein 90 (Hsp90), a chaperone implicated in PTSD [113,114]. When AhR detaches from HSP90, it enters the nucleus, where it binds to AhR nuclear translocator (ARNT). This biochemical reaction is opposed by the aryl hydrocarbon receptor repressor (AhRR), an inhibitor of AhR signaling [115]. If not inhibited by AhRR, the AhR/ARNT heterodimer enters the genome, where it activates the DNA replication-related element (DRE), initiating the transcription of many genes, including IL22 [116–118].

The effects of AhR are dependent on the attached ligands and are tissue- and context-specific; thus, it may exert opposite effects in one cell type vs. another. For example, activated AhR blocks cellular senescence in hepatocytes and fibroblasts, while in renal cells, it precipitates this phenotype [119,120]. Another transcription factor, HIF-1α, averts premature EC senescence by opposing the effects of AhR on microvascular endothelia [121]. Moreover, HIF-1α also protects enteric neurons as hypoxia caused by IEC oxygen consumption exhibits neuroprotective properties [122]. In another article, we have discussed HIF-1α lactylation and dysfunction in PTSD pathogenesis so we will not dwell on this topic here [123].

### 3.2. PTSD, Premature EC Senescence, and Barrier Function

It is well established that psychosocial stress activates the hypothalamic–pituitary–adrenal axis (HPA), and when the activation is chronic, it may precipitate PTSD. The initial step in HPA activation, is CRH synthesis in the paraventricular nucleus of the hypothalamus, a process mediated by AhR via acyloxyacyl hydrolase (AOAH). AOAH is an LPS-inactivating enzyme capable of neutralizing the translocated endotoxin, thus preventing tissue damage. We surmise that low AOAH levels reflect PTSD severity and can serve as an

early detection marker [109,124,125]. Another mechanism capable of neutralizing translocated bacteria or their components involves α7nAChRs and the CAP (see below) [126–128] (Figure 1). The CAP prevents premature vascular aging by enhancing the biosynthesis of nitric oxide (NO) in ECs, lowering the effects of psychosocial stress [129,130].

The question of why, under a similar level of trauma exposure, an individual develops PTSD while another does not, remains to be answered. However, as mentioned above, external (i.e., microbiome alterations) and internal (i.e., epigenetic changes) factors are likely involved. For example, psychosocial stressors have been associated with altered DNA methylation in epithelial and ECs, inducing senescence and increased gut barrier permeability [131–135]. Moreover, in ECs, impaired methylation of nuclear receptor subfamily 3 group C member 1 (NR3C1), a glucocorticoid receptor encoding gene, was demonstrated in both PTSD and cardiovascular disease (CVD), linking these conditions to premature endothelial aging [136–138]. In addition, AhRR methylation can promote PTSD and CVD via AhR/ARNT-induced EC senescence [139,140].

Recent studies have documented that AhR and HIF-1α exhibit opposing actions on ECs, as the former promotes while the latter opposes cellular senescence [111,126] (Figure 3). For example, AhR facilitates DNA methylation, contributing to EC senescence, a phenotype that can also be induced by disabling HIF-1α [5,141,142].

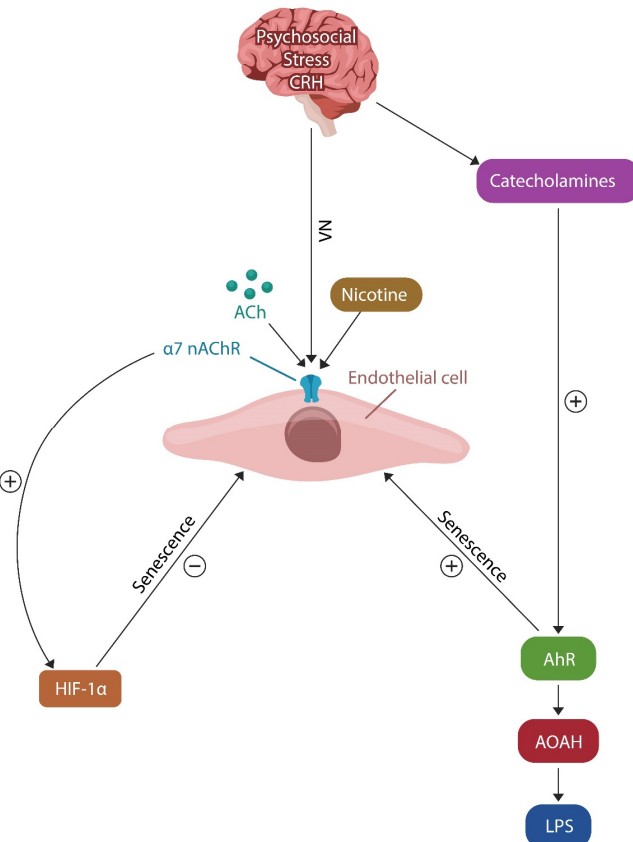

**Figure 3.** Psychosocial stress (via HPA and CRH) induces EC senescence by upregulating catecholamines. This action is opposed by VN via ACh or nicotine signaling with α7nAChR, upregulating HIF-1α and counteracting EC senescence. Psychosocial-stress-induced catecholamines upregulate AhR, inhibiting AOAH and facilitating LPS accumulation. Both AhR and LPS promote EC senescence.

## 4. PTSD Resilience Systems

Besides epigenetic mechanisms, psychosocial stress promotes EC senescence via four pathways: 1. AhR activation, 2. HIF-1α inhibition, 3. IL22 downregulation, and 4. disrupting tight junction molecules. These pathways enhance MT and the susceptibility for PTSD, as evidenced by previous findings:

1.  MT markers are elevated in other disorders characterized by dysfunctional gut barrier, including HIV, IBD, and schizophrenia, conditions highly comorbid with PTSD [143–145].
2.  Microbial cell-free DNA (mcfDNA) has been identified as a novel marker of MT and, to our knowledge, has not been evaluated for PTSD; however, it is utilized as a marker of sepsis, another condition characterized by MT [46,146,147].
3.  Lowered AOAH allows unopposed translocation of LPS into the host circulation and may serve as a PTSD susceptibility marker [108].
4.  Psychological stress downregulates intestinal IL22, increasing barrier permeability, suggesting that low levels of this cytokine may reflect PTSD susceptibility [148].
5.  Preclinical studies have implicated dysfunctional tight junction proteins, claudins and occludins, in PTSD, suggesting that these biomolecules can serve as markers of this pathology [149–151].

MT drew the attention of researchers and clinicians in the 1980s, during the HIV epidemic, as this virus, known for depleting IL22, has been associated with the disruption of the intestinal barrier [152]. This is significant as recent studies have found increased prevalence of mental illness in the offspring of mothers suffering from MDD during pregnancy. Since these pregnancies were associated with elevated levels of MT markers, these molecules may reflect psychopathology risk before birth [153–156]. Therefore, testing for MT biomarkers should be part of prenatal screening for mental illness in offspring.

### 4.1. Resilience Mechanism #1: The Cholinergic Anti-Inflammatory Pathway (CAP)

Intestinal macrophages are innate immune cells that recognize gut pathogens and clear both acute and chronic inflammation. These cells also regulate mucus secretion and intestinal motility, participating actively in the barrier function [157]. Macrophages express α7nAChRs and are directly innervated by the VN, averting premature epithelial and EC senescence [158] (Figure 4).

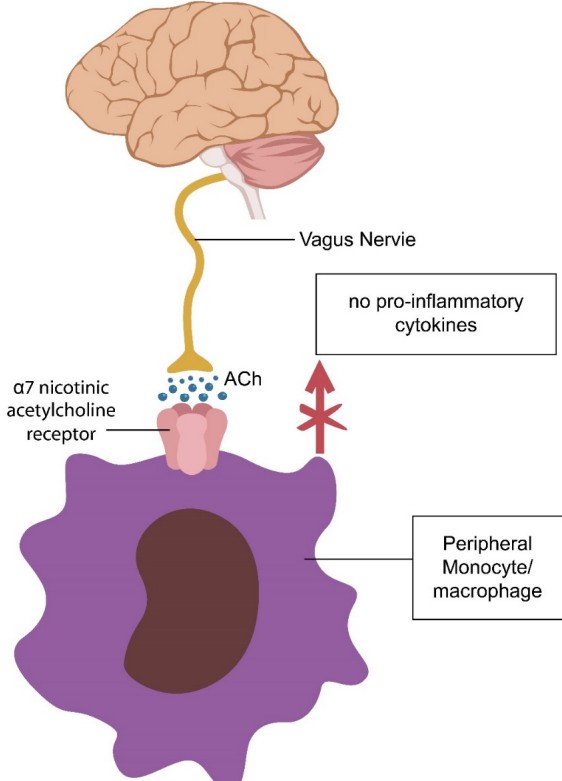

**Figure 4.** The VN communicates with the IC and comprises the CAP. Macrophages/monocytes, like ECs and IECs, express α7nAChR and are innervated by the VN, inhibiting the release of proinflammatory cytokines. The CAP is part of the interoceptive system as well as the gut–brain axis.

Due to the rapid turnover of enteric neurons and accelerated efferocytosis, intestinal macrophages are much more active compared to those in other tissues [159,160]. For this reason, dysfunctional elimination of dead cells and accumulation of molecular debris can trigger gut barrier disruption and inflammation. Conversely, taVNS may restore the integrity of the gut barrier, lowering the odds of PTSD [161–164] (Figure 4). Moreover, as mentioned above, IECs express α7nAChRs and are subject to the CAP, which protects the barrier function [165,166]. This is significant as it likely explains the high comorbidity of PTSD with IBD and HIV, disorders characterized by gut barrier dysfunction [25,88]. In addition, AOAH and IL22 protect the intestinal barrier against microbial and LPS translocation, indicating that these proteins could be used to identify PTSD early [124,167].

Dysfunctional CAP and macrophage-mediated efferocytosis can cause accumulation of cellular and molecular debris and gut barrier disruption [168].

### 4.2. Resilience Mechanism #2: HIF-1α and PTSD Resilience

HIF-1α is a transcription factor activated by local hypoxia that plays a key role in averting premature cellular senescence, including that of ECs. Senescence is a cellular program of averting malignant transformation by inducing cell entry into a state of permanent proliferation arrest, while maintaining an active metabolism. Although protective against cancer, the senescent phenotype releases a detrimental secretome, known as senescence-associated secretory phenotype (SASAP), which can disseminate senescence to the healthy neighboring cells [169]. Replicative cellular senescence was discovered by Leonard Hayflick in 1965, who demonstrated that cells do not replicate indefinitely, but after 40–60 divisions, they undergo telomere erosion and permanently exit the cell cycle [170,171].

Many neuropsychiatric disorders, including PTSD, have been associated with accelerated aging and shorter telomeres, indicating that psychosocial stress can induce premature cellular senescence [172,173]. Early endothelial aging in PTSD has been documented in many studies, suggesting that psychosocial stress primarily affects the endothelia and epithelia, contributing to increased intestinal permeability [174–176]. For example, MT has been documented in several PTSD studies, linking this pathology to an impaired gut barrier [177]. HIF-1α opposes MT via telomere elongation, reversing the effects of premature vascular aging, thus lowering the odds of PTSD [178,179].

As opposed to peripheral arteries in which HIF-1α rejuvenates endothelia, in the lungs, hypoxia causes pulmonary arterioles to constrict, leading to pulmonary artery resistance, which, under chronic conditions, may lead to right ventricular failure [180–182]. This may explain the high prevalence of obstructive sleep apnea (OSA) in PTSD patients, linking chronic hypoxia to this pathology [183,184]. Indeed, chronic HIF-1α elevation is detrimental, while intermittent hypoxemia is physiological and protects the endothelial barrier, suggesting that hyperbaric oxygen may comprise an effective treatment strategy for both PTSD and OSA. Indeed, several recent studies have reported amelioration of PTSD symptoms with hyperbaric oxygen, suggesting that more studies are needed to clarify this issue [185,186].

Aside from the endothelial barrier, HIF-1α also protects the intestinal epithelial barrier [187,188]. The mechanism of hypoxia-induced gut barrier repair is complex, but it comes down to IL22 upregulation. Microbiota-derived short-chain fatty acids (SCFAs) induce a degree of GI tract hypoxia due to increased oxygen consumption by the IECs. This, in turn, upregulates HIF-1α in mucosal ILC3, increasing the production of IL22 [189,190].

Although HIF-1α expression can be easily measured in blood serum, due to its fluctuating levels, it may not be an ideal PTSD marker; therefore, IL22, which indirectly reflects HIF-1α status, is likely a more reliable indicator of gut barrier integrity.

Taken together, PTSD association with gut barrier dysfunction is driven by the premature senescence of endothelial and epithelial cells. MT markers can be used to identify PTSD susceptibility prior to the development of clinical symptoms and should be included in prenatal care to screen for offspring mental health.

## 5. Early Detection Markers

In this section, we take a closer look at individual markers that could be used to identify PTSD in the premorbid phase, allowing prevention. Many markers of intestinal permeability have been utilized; however, at present, there is no gold standard test for MT [191]. Aside from the existing peripheral blood bacteria-detecting assays, including LPS, LBP, CD14, or AOAH, we propose two new tests, mcfDNA (measured via Karius Test®), and IL22 (measured via Singulex-Erenna®), which, when combined, may have higher specificity for MT. Indeed, we chose this battery of microbial tests as they measure both LPS translocation (LBP, CD14, and AOAH) and barrier integrity. LPS was chosen because of its predictive value for neuropathology. For example, human and animal studies show that maternal exposure to LPS during pregnancy contributes to high susceptibility for anxiety disorders, including PTSD, in offspring [192–196] (Table 1).

### 5.1. LPS

LPS is a component of the outer membrane of Gram-negative bacteria that consists of lipid A, the O antigen, and the core oligosaccharide [197]. Circulatory LPS originates in the GI tract, although lower amounts can be derived from the oral microbiome, and even food [198,199]. LPS is an inducer of host inflammatory responses and induces the release of IL1 alpha, IL1 beta, IL6, IL10, tumor necrosis factor alpha (TNF-alpha), and cyclooxygenase-2 as well as nitric oxide synthase (NOS), an enzyme that metabolizes arginine into nitric oxide (NO) and citrulline [200]. Interestingly, citrulline is an established marker of gut barrier integrity as well as an enhancer of athletic performance, suggesting anti-inflammatory properties [201,202]. In contrast, as a posttranslational modification, citrullination (due to excess citrulline) is detrimental as it promotes inflammation and autoimmunity [203,204].

Upon crossing into the host tissues, LPS is detected by Toll-like receptor 4 (TLR4), a protein that activates nuclear factor kappa B (NF-κB) and interferon regulatory factor 3 (IRF3), promoting the release of proinflammatory cytokines [205]. Minute amounts of LPS can lead to PTSD as this toxin (the lethal dose for humans is 1 to 2 μg) stimulates host immunity, inducing premature cellular senescence [206–209].

Psychosocial trauma can also activate TLR4, a process known as sterile inflammation, suggesting that the body does not discern very well between biological and psychosocial stress [210,211]. This may be significant as various disorders of unclear etiology, usually labeled as "psychogenic" or "functional", such as fibromyalgia or psychogenic fever, may be explained by the concept of sterile inflammation [212]. Conversely, psychotherapy, antidepressant drugs, and exercise have been demonstrated to lower TLR4 [213–215].

Sterile inflammation engenders pathology via TLR4-induced assembly of inflammasome (NOD)-like receptor protein 3 (NLRP3), which can be activated by LPS, psychosocial stress, a high-fat diet, and other environmental factors, including air pollution [216–218]. Excessive NLRP3 activation has been documented in PTSD, while inhibition of this inflammasome can ameliorate the symptoms, likely through AhR [219–221] (Figure 5). In addition, LPS has been reported to directly induce EC senescence in biological barriers, promoting MT [209,222].

Dysfunctional NLRP3 was demonstrated to aberrantly activate microglia, triggering the elimination of healthy neurons, a pathology manifested in PTSD as gray matter loss [223–225]. Indeed, neuroimaging studies have revealed decreased gray matter in PTSD, suggesting abnormal microglial activation [226–228].

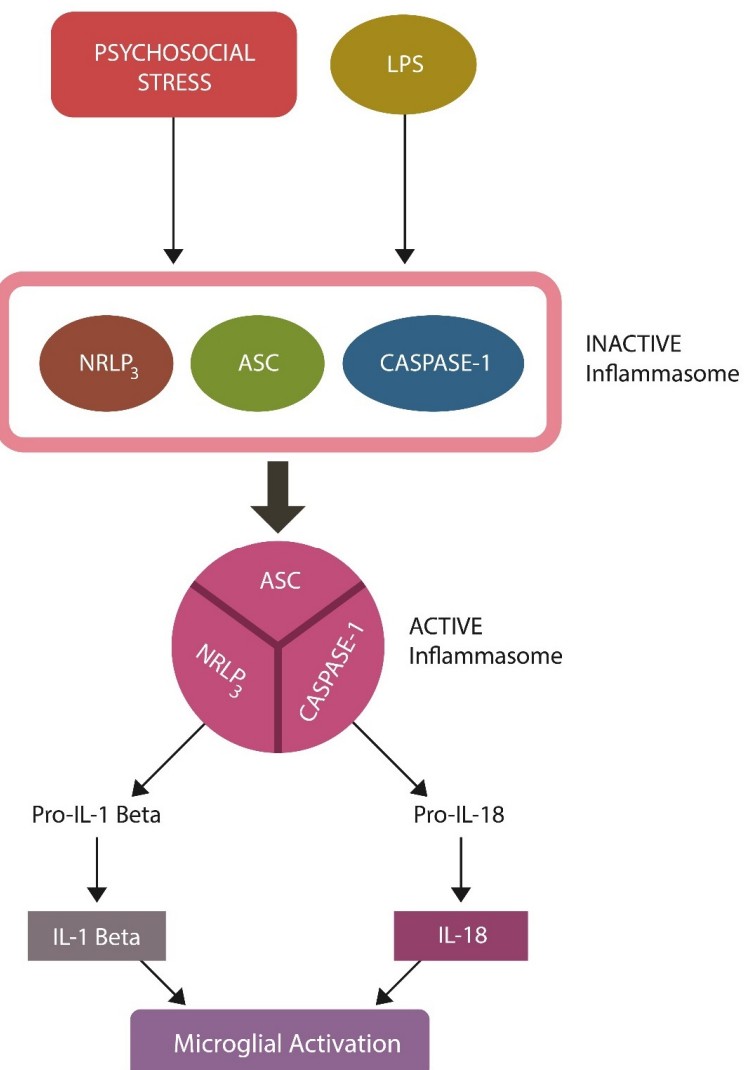

**Figure 5.** NLRP3 inflammasome activation: psychosocial stress and LPS trigger inflammation by inducing inflammasome assembly. Inflammasomes comprise a sensor, the nucleotide-binding oligomerization domain (NOD), in this case the (NOD)-like receptor protein 3 (NLRP3), apoptosis-associated speck-like protein containing a caspase recruitment domain (ASC), and caspase-1. In the presence of proinflammatory endogenous or exogenous stimuli, these molecules assemble and the inflammasome becomes biologically active, transforming premature IL-1 beta and IL-18 into their biologically active forms, which, in turn, activates microglia.

### 5.2. LBP and CD14

Upon LPS migration from the gut into the host systemic circulation, it binds LBP, a secretory phase-one molecule, as well as CD14, the receptor of the LPS-LPB complex [229]. Together, LBP/LPS/CD14 initiate cellular secretory responses that are much stronger than those of LPS alone, suggesting that these translocation markers enhance the pathological effects of translocated microbes or their components [230]. LBP serum levels are upregulated (normal range 5 to 10 μg/mL) and elevated LPB binds LPS with higher affinity, exacerbating PTSD [226,231]. Conversely, low LBP attenuates host inflammation, lowering the susceptibility for PTSD. In this regard, psychosocial stress has been demonstrated to upregulate LBP, inducing depressive symptoms and PTSD exacerbations, thus validating the role of the LBP/LPS/CD14 complex in these pathologies [232]. Conversely, the soluble form of CD14 (sCD14) is a reliable microbial translocation marker upregulated in sepsis, schizophrenia, and cancer, indicating that it likely comprises a valid signature of PTSD susceptibility [143,233,234].

### 5.3. Microbial Cell-Free DNA (mcfDNA)

Microbial cell-free DNA (mcfDNA) is a novel plasma marker utilized in infectious diseases that may also have potential to be used to detect translocated bacteria in PTSD [235]. The mcfDNA test was developed by Karius Inc. (Redwood City, CA 94065, USA) in 2017 (Karius Test®) and may accurately detect microbial translocation in PTSD [236]. To our knowledge, Karius Test® has never been used in PTSD; however, mcfDNA could serve as an early detection marker for this pathology. Further studies are needed to assess the validity of this marker in PTSD.

### 5.4. Acyloxyacyl Hydrolase (AOAH)

AOAH is a leukocyte-secreted lipase that hydrolyzes LPS by breaking the acyloxyacyl bonds [124]. For example, mice with higher AOAH levels were found to recover more rapidly from Gram-negative infections, while AOAH-deficient rodents exhibited not only more severe infection but also chronic pain, anxiety, and depression-like behaviors [109]. Interestingly, it was recently reported that AOAH regulates CRH via AhR, implicating this enzyme in PTSD pathogenesis [109]. In this regard, as AOAH eliminates LPS, low levels of this protein likely reflect PTSD vulnerability. Indeed, preclinical studies have shown that genetic treatment with the AOAH gene lowers LPS, suggesting a potential therapeutic strategy for Gram-negative sepsis and likely PTSD [237].

### 5.5. IL22

IL22 is a member of the IL-10 family of cytokines that is synthesized by several lymphocyte types, including T helper (Th) 17 cells, $\gamma\delta$ T cells, NKCs, and ILCs3 [238]. IL22 functions as a master regulator of gut barrier permeability as it modulates gut microbiota as well as the function of intestinal mucosa ILCs [239]. Luminal IL22 is generated by ILC3 and functions to enhance the beneficial gut microbes *Bifidobacterium* and *Lactobacillus* spp., increasing PTSD resilience. Along this line, we have suggested recombinant human IL22 as a treatment for schizophrenia and believe that this cytokine can also be beneficial in PTSD [240,241]. Indeed, psychological stress was demonstrated to disrupt IL22 as well as the gut barrier function, indicating that exogenous IL22 could restore the integrity of the intestinal barrier [148]. IL22 serum levels can be measured via Singulex-Erenna®, an ultrasensitive assay, indicating that detection of this cytokine is currently possible [242].

Taken together, based on several clinical and preclinical studies, PTSD susceptibility may be measured using MT markers, including elevated LPS, LBP, CD14, and lowered AOAH. Downregulated IL22, a marker of gut barrier integrity, likely reflects PTSD vulnerability. If validated, these biomarkers could be useful for discerning individuals at high risk of PTSD and adjusting or developing deployment policies accordingly.

**Table 1.** Microbial markers, sources, and physiological role.

| Markers | Origin or Function | References |
|---|---|---|
| HRV | GI tract inflammation lowers HRV | [68] |
| BP | Right vs. left IC = vasopressor vs. vasodilation | [79] |
| LPS (elevated) | Gut, oral microbiome, food origin | [198–200] |
| sCD14 (elevated) | Receptor for LPS/LBP heterodimer | [229] |
| LBP (elevated) | Enhances the action of LPS | [230] |
| AOAH (low) | Neutralizes translocated LPS | [124] |
| IL22 (low) | Impaired barrier function | [239] |

## 6. Conclusions

PTSD has accompanied human history for millennia. However, under a similar level of trauma exposure, who develops and who does not develop this condition remains poorly defined.

Shifting the research focus from the amygdala and fear to the IC and interoceptive awareness has brought forth new potential treatments as well as diagnostic opportunities, suggesting that dysgnosia may be corrected by interventions at the body periphery, such as taVNS. Since the IC is in close communication with both the heart and the GI tract, cardiac and gut biomarkers can be utilized for identifying PTSD susceptibility. For example, high-risk phenotypes, offspring of mothers with a history of MDD and bacterial (or viral) infections during pregnancy, may be identified early through postnatal screening.

This article emphasizes the prevalence of PTSD after war trauma, an aspect that may appear relevant only to military personnel, the police force, and their families. However, since a significant number of people from all walks of life are exposed daily to overwhelming stressors, including assaults, motor vehicle accidents, shootings, and hold-ups, this report may have significance for PTSD in general.

The battery of markers proposed here reflect intestinal barrier permeability (IL22) and translocation of microbial components (LPS, LBP, CD14, and AOAH). As they also mirror the BBB integrity, these biomarkers likely reveal the presence of bacteria or LPS in the CNS, including the IC.

**Author Contributions:** Conceptualization: A.S. and C.O.; writing and original draft preparation: Z.K.; Data curation and writing: C.K.; writing review: H.I.; Project administration J.J.A. All authors have read and agreed to the published version of the manuscript.

**Funding:** This research received no external funding.

**Conflicts of Interest:** The authors declare no conflict of interest. Where authors are identified as personnel of the International Agency for Research on Cancer/WHO, the authors alone are responsible for the views expressed in this article and they do not necessarily represent the decisions, policy or views of the International Agency for Research on Cancer/WHO.

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
