# Peer review of "Of Soldiers and Their Ghosts: Are We Ready for a Review of PTSD Evidence?"

_2673-8430, doi:10.3390/biomed3040039_

Round 1

Reviewer 1 Report

Comments and Suggestions for Authors

Overall, it is a very well written perspective. Here are some suggestions.

- The term “Hypertension as a PTSD risk factor” might be somewhat misleading or at least controversial. It might be more prudent to use a more neutral term on the direction of causal influences between PTSD and CVD including hypertension. In fact, some recent studies (e.g., PMID: 36069022) suggest the direction of influence is from PTSD to CVD, not the other way around. Hence, the implication in the use of term ‘risk factor’ might be misleading. 

- There are repeated definition of acronyms. For example, “hypertension (HTN)” is defined on line 90 and line 182. 

- Acronyms should be defined in their first occurrence. For example, “PTSD” was not defined in its first use.

- Use of more nuanced tone might be appropriate. For example, in line 272 and line 490, the use of the term “under similar circumstances” is too vague. Can you instead consider a term like “under a similar level of trauma exposure”. One can argue if all circumstances (environment, genetic factors, ..) are similar, how would the outcome change? Perhaps not in a deterministic fashion but it is not accurate to imply the outcomes are completely random. 

Reviewer 2 Report

Comments and Suggestions for Authors

I thank the authors and the editors for the opportunity to review this interesting paper, which styles itself as a narrative review of the literature on an alternate pathophysiology of PTSD involving a primary role for the insular cortex.

Despite boasting an extensive and seemingly comprehensive list of references that may imply this objective has been accomplished, the paper regrettably often strays from this mission. The authors draw from a range of often loosely related supporting evidence to advance a thesis that, at this stage, remains highly speculative.

The authors' desire to stimulate dialogue around this alternative hypothesis is certainly commendable. However, this study's dependence on generous and broad interpretations of citations to buttress its claims, coupled with its frequent overemphasis on the evidential support these provide, fatally undermines the credibility of their endeavor.

Consequently, this work should be primarily viewed as an opinion piece rather than as a scientific study, and should be clearly identified as such. In the present context, it would more appropriately be described as a hypothesis paper, rather than as a formal review.

Unfortunately, there are simply far too many instances where the authors indulge in or resort to speculative interpretation of the literature to bolster their thesis, to enumerate each case separately. However, for purposes of demonstration, an example of the concerns above are listed below for the paper's first two sections:

Page 1, line 29: References 2 and 3 are not appropriate citations to support the claim made in the second sentence of the introduction, which is ultimately speculative. 

Page 1, line 32: The citation of the Ilead is not appropriate in this context. The Ilead cannot be credibly cited to claim that Achilles was suffering from PTSD. Unsurprinsingly in war, as in life, violence simply begets violence, e.g. lex talionis. 

Page 2, line 60: Reference 13 does not support the claim made that “a growing number of clinicians” have made this observation. 

Page 2, line 79: Extraordinary claims require extraordinary evidence. References 25 and 26 do not support the fairly remarkable claim that the proposed mechanism “probably” explains “the high comorbidity between PTSD and inflammatory bowel disease”. 

Page 2, line 82: The precise meaning of “these findings” being “validated” by references 27 and 28 is not clear. These results do not clearly support, for example, the extraordinary claim above that PTSD and IBD are comorbid, because the “IC is tightly connected to the GI tract”, as is claimed. 

Page 2, line 88: The authors’ desire for a higher h-index notwithstanding, there does not appear to be any significant relevance of references 30 and 31 to the present work. 

Page 3, line 120: The discussion here of SARS and HIV infection appears entirely superfluous. The cited references 51 through 53 do not clearly support the claim made regarding these conditions disrupting insight.

Page 3, line 125: The claim being made here is that the gut likely serves a role in interoceptive awareness. References 26, 27, and 54 do not provide direct support for this remarkable claim. 

Page 3, line 133: The claims made here that because “many patients with PTSD prefer mindfulness”, “dysfunctional IC may drive … PTSD pathogenesis” is not supported by reference 59. Only the claim that some patients prefer these therapies is supported. The citations should therefore apply to the initial clause, not the final concluding clause.

Page 3, line 136: Reference 13 appears to support the claim made in the initial clause of this sentence, not the latter concluding clause.

Beyond these significant concerns, this study is considerably impaired by profound difficulties concerning its organization and structure. These difficulties primarily emerge from a discernible lack of precision and lucidity in outlining its objectives. The paper's intentions are communicated in two separate instances: once in the conclusion of the introduction, where the authors suggest the paper emphasizes "PTSD risk factors, heart rate variability... and hypertension... as well as several markers for early detection and prevention"; and again in the closing statement of the section titled "PTSD and interoceptive awareness", where the authors propose to explore "the role of IC in emotional intelligence, heart, and gut connection, along with PTSD risk factors, early detection indicators, and public health strategies for prevention".

Considering the unconventional premise of the authors, it would be reasonable for any single one of these topics to serve as the central focus for an individual paper. The authors' endeavor to weave all of these complex issues into a single narrative results in a paper that, regrettably, becomes too diffused, too fragmented, and too disarrayed. Consequently, the work presents itself as possessing only superficial depth and credibility, thus fatally undermining its overall impact.

Comments on the Quality of English Language

The paper could benefit from significant editing to ensure appropriate grammar throughout. 

Reviewer 3 Report

Comments and Suggestions for Authors

Paper overemphasizes the prevalence of war to average citizens. Even in periods when states are in conflict the numbers of citizens exposed to combat is relatively low, This should be recognized while not downplaying the trauma experienced by combat veterans. 

Reviewer 4 Report

Comments and Suggestions for Authors

This manuscript was very interesting! The connection of markers and gut health linked to PTSD was well presented.

The manuscript provides a wealth of summary information and pertinent references. The table at the end is clear.

I think the manuscript should be published as is. The discussion it will generate is needed.

Do the authors have any conflicts with the companies that would be able to test for MT? Please clarify.

Round 2

Reviewer 2 Report

Comments and Suggestions for Authors

I thank the author for these comments. However, these are regrettably insufficient to address the major concerns expressed in my original review. Although I share many of the author's concerns, and am sympathetic to the position expressed in the author's rebuttal, this cannot justify excusing serious concerns with the scientific quality of the submission.

Comments on the Quality of English Language

The quality could be improved in places. See, for example, this poorly-worded sentence in the abstract: "This tendency parallels PTSD research shift from amygdala and fear to the insular cortex and interoceptive awareness". 

Author Response

Please see attached responses
